# Medicinal Fungi with Antiviral Effect

**DOI:** 10.3390/molecules27144457

**Published:** 2022-07-12

**Authors:** Yu Zhang, Guoying Zhang, Jianya Ling

**Affiliations:** 1School of Pharmacy, Shandong University of Traditional Chinese Medicine, Jinan 250355, China; 2020110174@sdutcm.edu.cn; 2State Key Laboratory of Microbial Technology, Shandong University, Qingdao 266237, China

**Keywords:** medicinal fungi, antiviral effect, antiviral component

## Abstract

Pandemics from various viruses make natural organisms face challenges over and over again. Therefore, new antiviral drugs urgently need to be found to solve this problem. However, drug research and development is a very difficult task, and finding new antiviral compounds is desirable. A range of medicinal fungi such as *Ganoderma lucidum* and *Cordyceps sinensis* are widely used all over the world, and they can enhance human immunity and direct anti-virus activities and other aspects to play an antiviral role. Medicinal fungi are used as foods or as food supplements. In this review, the species of medicinal fungi with antiviral activity in recent decades and the mechanism of antiviral components were reviewed from the perspectives of human, animal, and plant viruses to provide a comprehensive theory based on better clinical utilization of medicinal fungi as antiviral agents.

## 1. Introduction

In today’s world, the outbreak of many infectious diseases has posed a serious threat to human life and health. As early as thousands of years ago, the outbreak of smallpox had caused tens of thousands of human deaths [1]. The outbreak of SARS-CoV-2 has now pushed mankind to the edge of the cliff. The outbreaks of most of these infectious diseases are related to viral infection. At this stage, the greatest threat to humanity is the RNA viruses, because they are very easy to mutate in the process of replication and reverse transcription [2]. This makes the prevention and treatment of RNA viruses very difficult. Classic antiviral drugs such as interferon and ribavirin are effective against most viruses in vitro, but usually not for patients, and because of a series of problems such as drug resistance and the high price of existing medicine [3,4]. More suitable antiviral drugs need to be found quickly.

Medicinal fungi are widely used all over the world. In Pakistan, nearly 23 species of medicinal fungi are used to treat colds, cough, influenza, asthma and other diseases [5]. The United States has added *Ganoderma lucidum* to the American Herbal Pharmacopoeia and Treatment Program [6]. The medicinal use of medicinal fungi also has a long history in China, which was recorded in Shennong Ben Cao Jing as early as 2000 years ago [7]. It can be used to treat or prevent diseases and has significant biological activity and medicinal value. A series of substances such as proteins, polysaccharides, and alkaloids can be produced in the mycelium, fruiting body, or spores, which has health effects on the human body [8]. Its antiviral, anticancer, and anti-inflammatory effects have attracted more and more attention in recent years [9,10,11]. Because of the special properties of medicinal fungi, their toxicity is much less than that of chemical drugs. During these years, some fungi have gradually developed into medicinal and edible fungi. Medicinal fungi are gradually welcomed by people, greatly improving people’s acceptance.

There has been a good summary of antiviral polysaccharides from medicinal fungi [7]. In this article, we further summarized the macromolecular and small molecular antiviral components of medicinal fungi and further classified them according to human viruses, animal viruses, and plant viruses (Table 1 and Table 2). The purpose of this review is to provide a reference for the development of medicinal fungi in the field of antivirus and to facilitate the follow-up antiviral research.

## 2. Antiviral Activity

Medicinal fungi can fight viruses in a variety of ways (Figure 1). At present, most existing antiviral drugs are effective mainly through the inhibition of the replication of the viruses in the body. Because medicinal fungi contain complex components, the synergistic effect of multiple components on the viruses in the body will often have an unexpected effect (Figure 2). We systematically summarized the antiviral effects of 18 species (Figure 3) of medicinal fungi from all over the world on different viruses in recent decades, and we elucidate the isolation sources and antiviral potential of the active ingredients.

### 2.1. Human Viruses

#### 2.1.1. SARS-CoV-2

Novel coronavirus disease 2019 (COVID-19) is an enveloped positive-stranded single-stranded RNA coronavirus of the betacoronaviridae family [107]. The high mutation rate of the RNA virus brings a serious challenge to the fight against the virus. From the beginning of Alpha, Beta, and Delta to today’s Omicron, more highly infectious variants have emerged one after another, resulting in a more powerful and contagious virus, making it more difficult to detect and control [108]. Therefore, we urgently need to develop new antiviral drugs to deal with this disease.

In a cell-based anti-SARS-CoV-2 test, several polysaccharides containing L-fucose isolated and characterized from *Ganoderma lucidum* were tested. It was found that the L-fucose–containing polysaccharides fraction 3 (RF3) had excellent antiviral activity (2 μg/mL) and was still effective at 1280 times dilution without cytotoxicity [12]. The impairment of the ACE/ACE2 ratio in coronavirus diseases is related to the pathological mechanism of COVID-19. ACE inhibitory proteins can be isolated from different medicinal fungi, including *Ganoderma lucidum*, *Grifola frondosa*, *Agrocybe species*, *Auricularia auricula*, *Hericium erinaceus*, *Hypsizygus marmoreus*, *Pleurotus cystidiosus*, *P. eryngii*, *P. flabellatus*, *P. florida*, *P. sajor-caju*, *Schizophyllum commune*, *Tricholoma giganteum*, and *Volvariella volvaceae*. In addition to peptides and proteins, ACE inhibitory triterpenes were extracted from *Ganoderma lucidum*. The ACE inhibition of these mushrooms can indirectly restore the ACE/ACE2 ratio, thus providing COVID-19’s treatment effect [13]. Antcins, a steroid compound in Taiwan’s endemic medicinal fungus *Antrodia cinnamomea*, has ACE2 inhibitory effect and can be used as new anti-ACE2 agents to combat SARS-CoV-2 infection [14].

The polysaccharides contained in *Lentinus edodes* have various biological activities and may play a positive role in the treatment of COVID-19. AHCC, a polysaccharide extracted from *Lentinus edodes* mycelia, has therapeutic effects on different types of viral infectious diseases, such as West Nile virus, influenza virus, hepatitis virus, and human papillomavirus, which indicates that regulating the immune system plays a theoretical role in protecting the host from a respiratory infection. Although AHCC has never been tested against human coronavirus, it is supposed to be a potential substance for treating COVID-19 [15]. Another polysaccharide IHL and a commercial lentinan CL extracted from the fruiting bodies of *Lentinus edodes* has the potential to treat lung injury and significantly reduce the level of inflammation in the lungs. CL extract can reduce early apoptosis induced by oxidative stress, while IHL extract can inhibit late apoptosis. These two polysaccharides can alleviate the damage caused by virus infection in the form of nutritional and health products in the future [16]. In recent years, network pharmacology techniques, including molecular docking and molecular dynamics simulation, have been widely used in drug screening. Verma et al., studied the molecular interaction between cordycepin, a key component of *Cordyceps sinensis*, and novel coronavirus-related target protein by molecular dynamics simulation, and found that cordycepin had strong binding force with RBD domain of novel coronavirus spike protein. This suggests that cordycepin may be used as a pan coronavirus fusion inhibitor targeting spike proteins to limit the entry of the virus into the body [17].

#### 2.1.2. Influenza Virus (IV)

IV belong to Orthomyxoviridae and have segmented single-stranded RNA genomes, which are usually divided into A, B, and C types. Influenza A virus is the most popular virus in the world [109]. The outbreak of the H1N1 virus in 2009 is one of the most serious cases in recent years [110]. At present, most influenza viruses spread seasonally. The most effective way to treat the influenza virus is to be vaccinated. However, the strong mutation ability of the virus leads to the need for vaccination year by year, which causes great inconvenience to people [111].

Obi et al., obtained extract ME, MFs (MF1, MF2, and MF3) from the fruiting bodies of *Grifola frondosa*. Adding ME and MFs to infected cells can reduce the yield of the virus. This may be related to the fact that the extracts induce the production of cytokines such as TNF-α, which can inhibit virus growth in vitro [18]. The components in *Phellinus igniarius* also showed good antiviral activity. Two compounds, the phelligridins E and G were isolated from the methanol extract of *Phellinus igniarius* fruiting bodies, which showed neuraminidase inhibitory activity against recombinant H1N1, H3N2, and H5N1 in a dose-dependent manner [19]. An in vitro experiment of *Phellinus igniarius* aqueous extract PIW shows that it has inhibitory activity on 2009 pandemic H1N1, human H3N2, avian H9N2, and oseltamivir-resistant H1N1 viruses, and its mechanism may be related to interfering with the early replication process of the virus [20]. In another study, the anti-influenza virus activity of LEM extracted from *Lentinus edodes* mycelium in vivo and in vitro was determined. The results showed that LEM could inhibit the growth of influenza virus in vitro and play a role in the early stage when the virus entered the host cell. In vivo experiments showed that LEM could activate the immune response and inhibit virus infection through the type I IFN pathway [21].

A compound Cryptoporic acid E (CAE), extracted from *Cryptoporus volvatus*, could inhibit the replication of the influenza virus and reduce the RNA level of the influenza virus [22]. The water extract from *Cryptoporus volvatus* fruiting body is also effective against H1N1 infection. The results showed that it could reduce the level of virus in cells, which may be related to the targeted inhibition of virus entry into cells by its water extract. In animal studies, the extract can completely protect mice from virus infection at high doses [23].

*Cordyceps militaris* also has a good curative effect against H1N1 infection. An acidic polysaccharide APS was isolated from *Cordyceps militaris* cultured in germinated soybean. The experimental study in vivo showed that APS could significantly reduce the titer of the virus in the lungs of mice, enhance the immune level and reduce the mortality after intranasal administration [24]. Furthermore, *Cordyceps militaris* extract was tested against H1N1 virus in DBA/2 mice. The results showed that the extract could reduce the mortality of mice infected with H1N1, increase the levels of cytokines such as TNF-α and IFN-γ, and activate the immune system by promoting the expression of IL-12 [25]. Meanwhile, the antiviral activities of mycelia extract of 10 medicinal fungi against the H1N1 virus in vitro were studied. *Auuriporia aurea*, *Flammulina velutipes*, *Fomes fomentarius*, *Ganoderma lucidum*, *Lentinus edodes*, *Lyophyllum shimeji*, *Pleurotus eryngii*, *Pleurotus ostreatus*, *Schizophyllum commune*, and *Trametes versicolor*, studies have shown that 10 kinds of medicinal mushrooms all showed inhibitory effect on H1N1. *T. versicolor* has the strongest antiviral effect, and the treatment index is 324.67 [26].

Some medicinal fungi have a good performance in indirectly fighting H1N1. DDEA, a fatty acid in *Cordyceps sinensis*, can reduce the level of inflammation caused by infection with H1N1 and reduce the secretion of inflammatory factors in cells. The anti-inflammatory effect is achieved by regulating the innate immune signal pathway activated by TLR-3-, RIG-I- and type I interferon [27]. A new polysaccharide PCP-II was isolated from *Poria cocos*. PCP-II combined with H1N1 vaccine could significantly enhance the antibody level, promote the proliferation of B lymphocytes and enhance the humoral immunity produced by the vaccine [28]. Mixed polysaccharides (MPs) were extracted from shiitake mushroom, *Poria cocos*, ginger, and dried tangerine peel, after oral administration of MPs and injection of vaccine in mice. It significantly improved the ability of mice to resist viruses, reduced lung injury, and enhanced the level of cellular immunity and humoral immunity [29]. The antiviral activity of medicinal fungi may also be related to its trace elements. Wang et al., found that selenium, zinc and magnesium contained in medicinal fungi may play a direct or indirect role in anti-influenza virus [112].

#### 2.1.3. Enterovirus 71(EV71)

EV71, a single-stranded RNA virus of about 7500 nucleotides, belongs to the Enterovirus genus of the Picornaviridae family [113]. EV71 virus infection can cause hand, foot, and mouth disease (HFMD) as well as serious neurological diseases, posing a serious threat to children under six years of age [114].

A new heteropolysaccharide GFP1 was extracted and purified from *Grifola frondosa* mycelia. The results of cell experiments in vitro showed that GFP1 inhibited the cytopathic effect (CPE) induced by EV71 in a dose-dependent manner. GFP1 inhibited the synthesis of EV71 virus RNA in Vero cells in a dose-dependent manner. The expression of EV71 virus polyprotein was analyzed by western blot. The results showed that GFP1 inhibited the expression of viral VP1 in EV71 infected cells and reduced the apoptosis induced by the EV71 virus by reducing the level of caspase-3 activation [30]. In another experiment on vaccine adjuvants, a protein-binding polysaccharide PS-G was extracted and purified from the fruiting bodies of *Ganoderma lucidum*. In vivo, it was found that the combination of EV-A71 mucosal vaccine and PS-G could activate the immune response and enhance the potential immune cell response to combat virus infection [31]. Two triterpenoids, GLTA and GLTB isolated from *Ganoderma lucidum* also showed inhibit effect on EV71. It was found that GLTA and GLTB had antiviral activity against EV71 virus infection and had no cytotoxicity. The results of molecular docking and PCR show that these two anti-EV71 compounds may block the uncoating process in EV71 infection, thus inhibiting virus infection [32].

#### 2.1.4. Human Immunodeficiency Virus (HIV)

In the early 1980s, human immunodeficiency virus type 1 (HIV-1) retrovirus was identified as the pathogen of acquired immunodeficiency syndrome (AIDS) [115]. The genome of HIV consists of two identical positive strands of RNA, with a total length of about 9.7 kilobase pairs. Antiretroviral therapy (ART) for HIV-1 infection is the main treatment for AIDS at present, which significantly reduces the morbidity and mortality associated with HIV/AIDS, but the problem of drug resistance still needs to be solved [116].

A novel serine protease cordysobin was purified from the *Cordyceps sinensis* fruiting bodies. The protein has obvious inhibitory activity on HIV-1 reverse transcriptase. The IC50 value is 8.2 × 10^−3^ μM [33]. The anti-HIV activity was also found in the water extracts from different parts of wild *Cordyceps sinensis* in vitro. The results showed that the five kinds of water extracts had a dose-dependent inhibitory effect on the HIV-1 virus. Further studies showed that their antiviral effect may be related to the activity of HIV-1 reverse transcriptase [34]. Jiang et al., successfully constructed a screening system for Escherichia coli against HIV-1 reverse transcriptase. Three kinds of adenosine, L3a, L3b, and L3c were purified from *Cordyceps militaris*. By using the screening system, L3a and L3b showed a high inhibitory effect on HIV-1 reverse transcriptase [35]. In another study, a lectin was extracted from the fruiting bodies of *Cordyceps militaris* which could inhibit the activity of HIV-1 reverse transcriptase with an IC50 value of 10 μM [36].

Li et al., isolated a lectin from *Pleurotus citrinopileatus*. The lectin had an inhibitory effect on HIV-1 reverse transcriptase with an IC50 of 0.93 μM [37]. In addition, two ribosome inactivating proteins were isolated from the *Hypsizigus marmoreus* and *Flammulina velutipes* fruiting bodies. Both have HIV-1 reverse transcriptase inhibitory activity [38,39]. They also successively isolated and purified a hemolysin monomer with a molecular weight of 29 kDa and a ribonuclease with a molecular weight of 20 kDa from *Schizophyllum commune* fruiting bodies, both of which could inhibit the activity of HIV-1 reverse transcriptase. The IC50 is 1.8 and 65 mM respectively [40,41].

Two kinds of crude extracts NGCs and AGCs from *Ganoderma lucidum* and *Ganoderma lucidum* has antler-like fruiting bodies. It was found that both crude extracts could inhibit the attachment of human immunodeficiency virus (HIV-1) glycoprotein 120 (gp120) to cell cluster differentiation 4 (CD4), and AGCs had a higher inhibitory effect on HIV-1 gp120 than NGCs [42]. Not only the crude extract but the triterpenoids have been found to have good inhibition of HIV-1 reverse transcriptase activity in *Ganoderma lucidum*. A molecular docking study shows that ganoderic acid B has a good docking effect with HIV-1 protease. HIV-1 protease is a suitable target for ganoderic acid B in general. There is great potential for AIDS treatment based on this compound [43]. Five lanosterane-type triterpenes isolated from spores of *Ganoderma lucidum*, ganoderic acid β, lucidumol B, ganodermanondiol, ganodermanontriol, and ganolucidic acid A, can inhibit the activity of HIV-1 reverse transcriptase with IC50 value of 20, 50, 90, 70 and 70 μM [44]. Ganoderiol F and ganodermanontriol isolated from the methanol extract of *Ganoderma lucidum* fruiting bodies were found to have anti-HIV-1 activity with an inhibition concentration of 7.8 μg/mL. And ganoderic acid B, ganoderiol B, ganoderic acid C1, 3b-5a-dihydroxy-6b-methoxyergosta-7,22-diene, ganoderic acid a, ganoderic acid H, and ganoderiol A can inhibit HIV-1 reverse transcriptase activity with a 50% inhibitory concentration of 0.17–23 mM [45].

Furthermore, a polysaccharide peptide(PSP)extracted from *Coriolus versicolor* could inhibit the activity of HIV-1 reverse transcriptase with an IC50 of 6.25 μg/mL. Inhibition of the interaction between HIV-lgp120 and fixed CD4 receptor IC50 is 1.5 μM [46]. Further experiments show that its antiviral mechanism may be related to reducing viral replication and promoting the secretion of specific antiviral chemokines [47]. In the study of a common medicinal fungus *Lentinus edodes*, a protein Lentin and a laccase with a molecular weight of 67 kDa were isolated from the fruiting bodies of *Lentinus edodes* and had inhibitory activity against HIV-1 reverse transcriptase with IC50 values of 7.5 μM and 1.5 μM respectively [48,49]. In addition, EPS4, water-soluble lignin extracted from the mycelia of *Lentinus edodes*, EPS4 completely inhibited the cytopathic effect induced by HIV-l when the concentration was more than 10 μg/mL. The water extracts E-P-LEM and LEM can block the cytopathic effect and the expression of specific antigens induced by HIV [50,51].

#### 2.1.5. Human Papilloma Virus (HPV)

HPV is a DNA virus with a double-stranded, closed, circular genome and an unenveloped icosahedral capsid of about 8 kb. HPV infection can lead to a variety of cancers, including cervical cancer, penile cancer, anal cancer, vaginal cancer, vulvar cancer, and oropharyngeal cancer [117]. HPV vaccination and regular physical check-ups are the main ways to prevent HPV infection [118]. At present, there is no specific drug for HPV treatment, which is usually treated by interferon, so it is found that the specific drug for HPV is very interesting.

In a clinical trial, *Trametes versicolor* and *Ganoderma lucidum* fruiting bodies powder were put into capsules for oral HPV patients, and 87.8% of the patients were cured after treatment. This confirmed the curative effect of the two kinds of fungi and laid the foundation for further research [52]. In a preliminary study of asymptomatic women, the use of a vaginal gel based on *Trametes versicolor* for 12 days significantly improved the composition of vaginal microbial communities, helping to improve the state of cervical epithelium and vaginal health [53]. Another study shows that the use of gel to treat HPV-positive people is effective [119].

#### 2.1.6. Dengue Virus (DENV)

DENV is an arbovirus that mainly relies on Aedes aegypti mosquitoes as a vector to transmit to humans. It has four different serotypes: DENV-1, DENV-2, DENV-3, and DENV-4. Among them, DENV-2 is the most widely spread [120,121]. Dengue fever is a global public health threat, infecting 100 to 400 million people every year, with tropical and subtropical regions being the most affected [122].

The treatment of Aedes aegypti mosquitoes with *Beauveria bassiana* spores can significantly inhibit the replication of the dengue virus in the midgut of mosquitoes, which is partly related to the immune activation of effector genes controlled by Toll and JAK-STAT pathways [54]. Panya et al., found that cordycepin can inhibit the replication of DENV and reduce the RNA level of the virus in Vero cells. Its antiviral effect is related to the replication of virus RNA in the later stage of virus infection [55]. In another study, five kinds of medicinal fungi *Lignosus rhinocerotis*, *Pleurotus giganteus*, *Hericium erinaceus*, *Schizophyllum commune*, and *Ganoderma lucidium* were extracted with hot aqueous and ethanol, and then extracted with n-hexane, ethyl acetate and water in turn. The anti-dengue virus activity of the extract was evaluated by a plaque reduction test. The results showed that hot aqueous extracts and aqueous soluble extracts of *L. rhinocertis*, *P. giganteus*, *H. erinaceus*, and *S. commune* had the least toxicity to Vero cells and showed very significant anti-DENV-2 activity [56]. The therapeutic effects of the above extracts on dengue virus-induced inflammation were further studied. The results showed that hot aqueous extracts of *G. lucidium*, *S. commune*, *P. giganteus*, and aqueous soluble extracts of *L. rhinocerotis* successfully inhibited the production of cytokines in monocytes infected by dengue fever [57].

The inhibitory activity of *Ganoderma lucidum* extract on DENV NS2B-NS3 protease was determined in two other studies. The results showed that the inhibition rate of aqueous extract on DENV2 NS2B-NS3 protease was 84.6 ± 0.7%. The main component of the extract was hesperidin by LC-MS. It was proved by molecular docking and density functional theory analysis that hesperidin was an efficient inhibitor of NS2B-NS3 protease [58]. Bharadwaj et al., screened the inhibitory activity of triterpenes from *Ganoderma lucidum* on DENVNS2B-NS3 protease by molecular docking and obtained 22 kinds of triterpenoids such as Ganodermanontriol. Then the activity of the obtained components was evaluated in vitro, and the virus infection experiment in vitro showed that Ganodermanontriol was a potential bioactive triterpene [59].

#### 2.1.7. Hepatitis Viruses (HV)

At present, five kinds of hepatitis viruses are prevalent in the world. There is hepatitis A virus (HAV), hepatitis B virus (HBV), hepatitis C virus (HCV), hepatitis D virus (HDV), and hepatitis E virus (HEV). Except HBV is a DNA virus, all other viruses are RNA viruses. Hepatitis B and C caused by hepatitis virus are the two types with the highest mortality. If chronic hepatitis caused by virus infection cannot be treated in time, it will lead to very serious liver cancer.

##### Hepatitis B Virus (HBV)

Corbrin capsule prepared from *Cordyceps sinensis* extract can reduce the apoptosis of renal tubular epithelial cells induced by HBX protein caused by hepatitis B virus infection. The mechanism is to inhibit the apoptosis of HK-2 cells enhanced by HBX by inhibiting the PI3K/Akt/Bcl-2 pathway [60]. The GF-D extracted from *Grifola frondosa* by Gu et al., has an inhibitory effect on HBV, and has a synergistic effect with human interferon α-2b in cell experiments. The antiviral mechanism of GF-D may be that it directly interferes with HBV replication at the level of DNA polymerase [61]. In another study, a large number of clinical data show that *Polyporus umbellatus* polysaccharides (PUPS) alone or in combination with interferon and other drugs can treat hepatitis caused by HBV [62].

Moreover, a polysaccharide sizofiran (SPG) isolated from *Schizophyllum commune* can regulate the cellular and humoral immune response of nucleocapsid antigen in patients with chronic hepatitis B and improve the level of antibody [63]. Li et al., found that *Ganoderma lucidum* and the aqueous extract of *Radix Sophorae flavescentis* co-fermented broth can significantly enhance the anti-hepatitis B virus activity in vitro and enhance the hepatoprotective effect of the body to reduce liver injury compared with the simple mixture of these two ingredients [64]. Ganoderic acid also can inhibit intracellular HBV replication and reduce the activities of alanine aminotransferase and aspartate aminotransferase (ALT and AST) in serum to protect against liver injury [65]. Zhao et al., optimized the ultrasonic extraction process of lentinan. Two kinds of polysaccharides, LEP-1 and LEP-2, were extracted by an optimized ultrasonic extraction method. The in vitro study showed that both polysaccharides showed anti-hepatitis B virus activity [66].

##### Hepatitis C Virus (HCV)

Matsuhisa’s team found that *Lentinula edodes* mycelia solid culture extract (MSCE) and its main active component low-molecular-weight lignin (LM-lignin) could inhibit the entry of HCV pseudovirus (HCVpv) into cells, and LM-lignin inhibited HCVpv entry at a concentration lower than MSCE [67]. A kind of *Cordyceps militaris* (CM) capsule can inhibit HCV RNA replication in vitro. Further study showed that cordycepin is the main activity component of CM anti-HCV. The effect of cordycepin is related to its inhibition of NS5B polymerase activity [68].

#### 2.1.8. Herpes Viruses (HV)

HV is a kind of virus with an envelope and double-stranded DNA genome. HV can be divided into three types: α herpesvirus, β herpesvirus, and γ herpesvirus. Herpes simplex virus type 1 (HSV-1) and herpes simplex virus type 2 (HSV-2) are two popular viruses, and they both belong to the α herpes virus. HSV-1 mainly causes oral and facial infections, while HSV-2 is associated with genital herpes. Epstein-Barr virus (EBV) belongs to γ herpesvirus. Infection of EBV may lead to nasopharyngeal carcinoma (NPC) [123,124].

##### Herpes Simplex Virus (HSV)

A novel *Grifola frondosa* protein GFAHP can inhibit HSV-1 replication in vitro. The IC50 value is 4.1 μg/mL and the therapeutic index is more than 29.3. In vivo experiments show that GFAHP can alleviate the inflammatory response of mice and reduce the production of virus in vivo [69]. A *Lentinus edodes* mycelia extract JLS-S001 also can significantly inhibit the infection of HSV-1 to cells. This may be related to the fact that JLS-S001 blocks HSV-I replication at the later stage of the virus replication cycle [70]. In another study, an acidic protein-binding polysaccharide APBP was isolated from the fruiting bodies of *Ganoderma lucidum*. It has antiviral activity against both HSV-1 and HSV-2. The EC50 of APBP to HSV-1 and HSV-2 was 300 and 440 mg/mL. The antiviral activity of APBP may be mainly due to the inhibition of HSV attachment and penetration to Vero cells [71]. Further experiments were conducted to explore the efficacy of APBP combined with commonly used antiviral drugs. When APBP was combined with interferon, it showed a synergistic effect on the virus [72]. The results of the combination of APBP with acyclovir (ACV) and adenosine arabinoside (ara-A) showed that the combination of APBP, ACV, and ara-A showed a synergistic effect on HSV-1. On the other hand, APBP combined with ACV showed a synergistic effect on HSV-2, while combined with ara-A it showed an antagonistic effect on HSV-2 [73].

In addition, the crude extracts of various medicinal fungi also showed anti-HSV activity. Two kinds of water-soluble extracts, GLhw and GLlw, and eight kinds of methanol-soluble extracts, GLMe-1-8, were extracted from *Ganoderma lucidum*. The antiviral experiment in vitro showed that GLhw, GLMe-1, GLMe-2, GLMe-4, and GLMe-7 could significantly inhibit the cytopathic effect of HSV [74]. WTTCGE, an herbal mixture containing *Ganoderma lucidum*, can improve the symptoms of patients with genital and lip herpes and shorten the recovery time of patients [75]. Based on other recent experimental findings, the anti-HSV-1 virus activity of methanol extract and water extract of 10 kinds of medicinal fungi collected from Turkey was studied. The results showed that the water extract of *Fomes fomentarius*, *Phellinus igniarius*, and *Porodaedalea pini* showed strong anti-herpes activity [125]. The mycelia extract of 10 medicinal fungi were tested against the HSV-2 virus in vitro. It was found that four species of *Pleurotus ostreatus*, *Fomes fomentarius*, *Auriporia aurea*, and *Trametes versicolor* had an inhibitory effect on the virus. The highest treatment index of *T. versicolor* was 324.67 [27].

##### Epstein-Barr Virus (EBV)

Cordycepin can inhibit virus infection by affecting the synthesis of virus protein by acting on the gene of EBV, and resist the proliferation of tumor cells caused by EBV infection, and has certain anti-tumor activity [76]. Terpenoids in *Ganoderma lucidum* may be used as a potential EBV antigen inhibitor. Iwatsuki et al., studied the inhibitory effect of 17 terpenoids on EB virus early antigen (EBV-EA). The results showed that 16 compounds showed strong inhibitory effects on EBV-EA induction [77]. Zheng found that five triterpenoids can inhibit the activation of EBV antigen and telomerase activity through in vitro experiments and molecular docking studies. It is speculated that it has a therapeutic effect on NPC caused by EBV virus [78].

#### 2.1.9. Respiratory Syncytial Virus (RSV)

RSV is a single-stranded RNA virus, which mainly infects infants with weak immunity and can cause acute lower respiratory tract infection [126]. Since RSV infection does not lead to long-term immunity, repeated infections may occur in children and adults. Ribavirin is currently the only RSV antiviral drug licensed. At present, as the only licensed RSV antiviral drug, Ribavirin has some unsolved defects, such as high cost, uncertain curative effect, potential toxicity, etc. [127].

An immunomodulatory protein FIP-fve was isolated from *Flammulina velutipes*. In vivo experiments showed that oral administration of FIP-fve could significantly reduce the viral titers of RSV and the level of mRNA in the lungs of mice and reduce the secretion of IL-6. FIP-fve may inhibit RSV replication and RSV-induced inflammation by reducing NF-κB translocation [79].

#### 2.1.10. Poliovirus (PV)

Poliovirus is classified as an enterovirus within the Picornaviridae. The virus genome is a single-stranded (+)-stranded RNA with a length of about 7500 nucleotides [128]. PV usually invades the central nervous system after infection, resulting in permanent flaccid paralysis [129].

In a study, anti-PV experiments were carried out on aqueous extract AqE, ethanol extract EtOHE and lentinan LeP of *Lentinus edodes*. The results showed that the three extracts showed antiviral activity and mainly played an inhibitory role in the early stage of virus infection [80].

#### 2.1.11. Rabies Virus (RV)

RV is an RNA virus. Rabies caused by RV infection is a fatal and incurable encephalomyelitis. It can kill tens of thousands of people worldwide every year. RV is usually parasitic in many wild animals and can be transmitted through damaged skin and mucosa [130]. At present, the main treatment for rabies is vaccination, which can prevent symptoms and death after timely and effective treatment [131].

A polysaccharide PCP-II isolated from *Poria cocos* can be used as a good adjuvant for the rabies vaccine. PCP-II can enhance the level of rabies-specific humoral immunity and cellular immune response and improve the protective effect of vaccines [81].

#### 2.1.12. Marburg Virus (MARV)

MARV first appeared in 1967 and was first found in African green monkeys. MARV is one of the deadliest human pathogens in the world. MARV and Ebola virus (EBOV) belong to the filamentous family. The infection will cause high fever, diarrhea, vomiting, and other symptoms, and the mortality rate is very high. There is no effective drug for treatment at present [132].

The Gai team successfully constructed a MARV virus-like particle (VLP). When *Poria cocos* polysaccharide PCP-II was combined with MARV VLP, the PCP-II group significantly enhanced the specific antibody response and neutralization antibody titer of MARV VLP and improved the immune level of mice [82]. The protective effect of PCP-II combined with MARV VLP was further evaluated in primate rhesus monkeys. The results showed that MARV VLPs mixed with PCP-II had excellent immunogenicity in rhesus monkeys [83].

As far as medicinal fungi against human viruses are concerned, many substances with therapeutic, with antiviral effects or substances that cause some other antiviral effects have entered clinical research and commercial production. CL, a commercial lentinan with the potential to treat lung injury, is expected to be used as a health food to regulate lung injury caused by viral infection. The listed drug Corbrin capsule containing *Cordyceps sinensis* extract has a therapeutic effect on hepatitis B virus infection. A kind of *Cordyceps militaris* capsule can affect the replication of HCV. In clinical studies, capsules containing *Trametes versicolor* and *Ganoderma lucidum* fruiting body powder have therapeutic effects on patients with oral HPV. Vaginal gel based on *Ganoderma lucidum* can significantly improve the clinical symptoms of HPV positive patients. *Polyporus umbellatus* polysaccharides (PUPS) can treat hepatitis caused by HBV. *Schizophyllum commune* polysaccharide sizofiran (SPG) can improve the antibody level of patients with chronic hepatitis B. WTTCGE, an herbal mixture containing *Ganoderma lucidum*, can significantly improve the clinical symptoms of patients with herpes.

### 2.2. Animal Viruses

#### 2.2.1. Infectious Hematopoietic Necrosis Virus (IHNV)

IHNV is the pathogen of infectious hematopoietic necrosis (IHN), which mainly causes the infection of many salmon species. It seriously affects the aquaculture industry [133]. A lentinan LNT-I had significant antiviral activity against IHNV in vitro. The antiviral mechanism of LNT-I was related to direct inactivation and inhibition of virus replication [84].

#### 2.2.2. Muscovy Duck Reovirus (MDRV)

MDRV has high pathogenicity and can infect ducks, chickens, and other poultry. After infection, poultry will have watery diarrhea, leg weakness, and other symptoms. MDRV has had a serious impact on the poultry industry [134].

Polysaccharide HEP extracted from the fruiting bodies of *Hericium Erinaceus* can effectively relieve the clinical symptoms and reduce the mortality of diseased Muscovy duck. Further studies have shown that HEP can enhance the defense function of the intestinal mucosal immune system, improve the number of intestinal mucosal immune-related cells, and improve the secretion of intestinal sIgA, IFN-γ, and IL-4 to prevent and treat MDRV infection [85]. The transcriptome analysis of duodenal specimens showed that HEP could regulate the homing process of muscovy duck lymphocytes to resist MDRV infection [86]. HEP can also reduce the damage of immune organs caused by MDRV and reduce apoptosis [87].

#### 2.2.3. White Spot Syndrome Virus (WSSV)

WSSV is the cause of white spot disease (WSD), which mainly infects shrimp and many other crustaceans. The mortality rate is as high as 90–100% within three to seven days after infection. Currently, WSSV is considered to be the deadliest viral pathogen in shrimp and many other crustaceans [135]. A β-1,3 glucan (BG) was extracted from *Schizophyllum commune*. In the shrimp feeding experiment, BG was added to the feed of shrimp infected with WSSV. Compared with the control group, BG could enhance the immune level and improve the survival rate of shrimp [88].

#### 2.2.4. Feline Immunodeficiency Virus (FIV)

FIV is a virus that can cause immunodeficiency syndrome in cats. FIV is spread mainly through bites between cats. The mortality rate of cats infected with FIV is not high. At present, vaccination can effectively prevent FIV infection [136].

In a study, seventeen kinds of medicinal fungi were extracted with water, ethanol, and hexane, respectively, and the inhibitory effect of the extract on FIV reverse transcriptase (FIV-rt) was studied. The results showed that the ethanol extract of *Inonotus obliquus* had the best activity. The IC50 value is 0.80 ± 0.16 µg/mL. In addition, ethane extract of *Inonotus obliquus*, water and ethanol extract of *Phellinus igniarius*, ethanol extract of *Cordyceps sinensis* fruiting body, hexane extracts of *Inonotus obliquus* mycelium, ethanol extracts of *Ganoderma lucidum*, hexane extracts of *Morchella esculenta* fruiting bodies, and fresh fruiting bodies of *Cordyceps sinensis* also showed good activity [89]. Moreover, a kind of commercially available compound HELP-TH1 is primarily composed of *Ganoderma lucidum*, *Cordyceps sinensis*, and *Trametes versicolor*. The experimental results in vitro show that HELP-TH1 can reduce the FIV load [90].

#### 2.2.5. Deformable Wing Virus (DWV)and Lake Sinai Virus (LSV)

DWV and LSV are two common viruses that can infect honeybees, which can cause honeybee wings to shrink and can also shorten their lifespan [137]. Based on recent experimental findings, by adding *Fomes fomentarius* and *Ganoderma applanatum* extract to the feed of the honeybee population, the titer of DWV and LSV virus in honeybees could be significantly reduced in a dose-dependent manner [91].

#### 2.2.6. Nerve Necrosis Virus (NNV)

Viral nerve necrosis disease is caused by NNV infection, which is a disease with high infectivity and high mortality. NNV mainly infects fish such as grouper, among which larvae and juveniles are the main targets of infection. This has caused great economic losses to the aquaculture industry [138]. A protein rLZ-8 was extracted, expressed, and recombined from *Ganoderma lucidum*. The antiviral effect of rLZ-8 on infected NNV fish was verified by one experiment in vitro and three experiments in vivo. The results showed that rLZ-8 could activate the immune defense of fish and effectively fight against virus infection [92].

#### 2.2.7. Porcine Circovirus Type 2 (PCV-2)

PCV-2 is a widespread epidemic virus in pigs. Pigs infected with PVC-2 usually show symptoms after weaning, resulting in severe damage to the immune system. All kinds of secondary diseases seriously threaten the life of pigs [139].

Liu et al., inoculated mice with *Ganoderma lucidum* polysaccharide liposome Lip-PS and inactivated PCV-2. Compared with inoculation inactivated PCV-2 only, combined vaccination can induce a more effective specific immune response to PCV-II infection [93].

#### 2.2.8. Porcine Reproductive and Respiratory Syndrome Virus (PRRSV)

PRRSV is an enveloped positive-strand RNA virus which is a pathogen causing porcine reproductive and respiratory syndrome (PRRS). Pigs will develop reproductive disorders and respiratory symptoms after infection. At present, there is no specific cure for PRRSV infection, but the virus spreads in many ways and it is difficult to prevent it [140].

Two monomer compounds, C_M-H-L-5_ and 5α,8α-epidioxy-22E-ergosta-6,22-dien-3β-ol were isolated from the *Cryptoporus volvatus*. Through the antiviral experiment in vitro, the results showed that they had anti-PRRSV activity [94,95]. In another study, antiviral experiments were carried out on the water extract of *Cryptoporus volvatus* fruiting bodies in vivo and in vitro. The results showed that it had an effective antiviral effect on PRRSV infection. The extract mainly inhibited the entry of PRRSV and the synthesis of PRRSV RNA, and it could inhibit the replication of the virus in vivo. The survival rate of pigs was improved [96].

#### 2.2.9. Porcine Delta Coronavirus (PDCoV)

PDCoV was first detected in fecal samples of Asian pigs in 2009. PDCoV is a porcine intestinal coronavirus. Clinical symptoms of PDCoV infection include diarrhea, dehydration, vomiting, and the death of newborn piglets. There is no effective drug to treat it at present [141].

An ergosterol peroxide EP was extracted from *Cryptoporus volvatus*. The in vitro results showed that EP could effectively prevent the attachment and entry of PDCoV. PDCoV may activate p38/MAPK signal pathway to promote its replication, activate the NF-κB signal pathway to stimulate the expression of many cytokines, and cause inflammation. EP treatment can inhibit p38 and NF-κB activation induced by PDCoV infection [97]. In vivo, experimental studies showed that oral EP could reduce the pathological manifestations caused by PDCoV infection and reduce the viral load of piglets. It further explains the therapeutic effect of EP on the body by acting on p38/MAPK and NF-κB signal pathways [98].

#### 2.2.10. Bovine Herpesvirus 1 (BoHV-1)

BoHV-1, a DNA virus classified as an alpha herpes virus, is the main pathogen of cattle. Infectious bovine rhinotracheitis, abortion, infectious pustular vulvovaginitis, and other symptoms often occur in infected cattle [142]. Anti-BoHV-1 experiments were carried out on aqueous extract AqE, ethanol extract EtOHE and lentinan LeP of *Lentinus edodes*. The results showed that the three extracts showed antiviral activity and mainly played an inhibitory role in the early stage of virus infection [80].

#### 2.2.11. Newcastle Disease Virus (NDV)

NDV is the main cause of Newcastle disease (ND). NDV is usually parasitic in chickens. It was first discovered as early as 1926, with high mortality and high infectivity. At this stage, vaccination is the best choice to prevent NDV infection [143].

The anti-neuraminidase activity of NDV was tested with extracts from different organic soluble fractions of *Ganoderma lucidum*. The results showed that methanol and n-butanol fractions had stronger anti-neuraminidase activity, which may be related to the acidic pH value of *Ganoderma lucidum* extract and the effect of flavonoids on NDV neuraminidase activity [99]. *Ganoderma lucidum* polysaccharide GLP can significantly promote the proliferation of lymphocytes and increase the level of interferon-γ (IFN-α) mRNA in vitro. In vivo experiments showed that oral GLP could significantly promote lymphocyte proliferation and improve serum antibody titer in chickens. This shows that GLP can be used as a good adjuvant in combination with vaccines [100]. Moreover, three kinds of polysaccharides were extracted and purified from *Auricularia auricula*. After chlorosulfonic acid–pyridine method modification, their inhibitory effects on NDV were evaluated in vitro. Compared with the non-modified polysaccharides, the sulfated polysaccharides significantly enhanced their antiviral activity. This is related to the role of sulfated polysaccharides in the early, middle, and late stages of entering cells from the virus [101]. In an in vitro and in vivo study, two *Cordyceps militaris* polysaccharides CMP40 and CMP50 could stimulate the proliferation of lymphocytes and increase the titer of serum antibody and the levels of interferon and IL-4. It is suggested that CMP40 and CMP50 can significantly improve the immune effect of Newcastle disease vaccine and are expected to be used as a new type of immune adjuvant [102].

Medicinal fungi also have good prospects in the fight against animal viruses. A number of studies have shown that medicinal fungal extracts have good efficacy in animals. *Hericium erinaceus* polysaccharide (HEP) can effectively relieve the clinical symptoms and reduce the mortality of Muscovy ducks infected with MDRV. β-1,3 glucan (BG) in *Schizophyllum commune* can improve the immune level of shrimp to resist WSSV. *Fomes fomentarius* and *Ganoderma applanatum* extracts could significantly reduce the titer of DWV and LSV viruses in honeybees. The recombinant protein rLZ-8 in *Ganoderma lucidum* can activate the immune defense of fish and effectively combat virus infection. EP, a peroxidized ergosterol from the *Cryptoporus volvatus*, can reduce the pathological manifestations caused by PDCoV infection and reduce the viral load of piglets. *Ganoderma lucidum* polysaccharide GLP and *Cordyceps militaris* polysaccharide CMP40, CMP50 can significantly improve the immune effect of Newcastle disease vaccine. These substances are expected to be used as a food supplement or agents for veterinary practice.

### 2.3. Plant Viruses

#### 2.3.1. Tobacco Mosaic Virus (TMV)

TMV is a common plant RNA virus, which mainly infects tobacco, tomato, and other crops. Plants infected with TMV will appear with symptoms such as yellowing and the wrinkling of leaves, which cause great inconvenience to agricultural production [144].

Two new steroid compounds, leiwansterols A and B, and three known compounds were isolated from *Omphalia lapidescens*. The results of in vitro experiments showed that all the five compounds had anti-tobacco mosaic virus activity [103]. Furthermore, a polysaccharide BAS-F was extracted from *Fomes fomentarius*. After using BAS-F on tobacco leaves infected with TMV, it was found that BAS-F could completely inhibit leaf infection, and have no toxic effect on host plants [104]. Wang’s team successfully extracted lentinan LNT and modified it to get sulfated lentinan sLNT. Through the proliferation inhibition experiment, the results showed that both sLNT and LNT could inhibit virus replication, and the inhibitory effect of sLNT was stronger than that of LNT. Its mechanism may affect the affinity of the TMV coat protein to host and activate some defense genes [105].

#### 2.3.2. Groundnut Bud Necrosis Virus (GBNV)

GBNV can cause sprout blight on legumes, tomatoes, peppers, potatoes, cotton, and other crops, which seriously restricts the development of the planting industry [145].

The culture filtrate prepared from the mycelia extracts of *Coprinopsis cinerea*, *Ganoderma lucidum*, and *Lentinula edodes* was tested against GBNV in vitro. The results showed that spraying the mixed culture filtrate could reduce the number of pathological changes and the titer of the virus. Finally, through GC-MS analysis, it was found that Squalene, a triterpenoid in Ganoderma lucidum, may have a potential antiviral effect [106].

Although there are few studies on medicinal fungal extracts against plant viruses, several medicinal fungal extracts have good therapeutic effects on diseased plants in the existing studies. BAS-F, a polysaccharide in *Fomes fomentarius*, showed significant inhibitory effect on TMV and no toxic effect. The culture filtrate of mycelia extracts from *Coprinopsis cinerea*, *Ganoderma lucidum* and *Lentinula edodes* could significantly reduce the titer of GBNV in plants. All these can provide important reference significance for the use of medicinal fungi extract in agricultural production.

## 3. Conclusions

It can be seen from our review that medicinal fungi are not only crude extracts, but also that some single components, such as proteins, polysaccharides and terpenoids, can effectively combat viral infection. Some of these medicinal fungi also have inhibitory effects on a variety of viruses (Figure 4). The complex components are closely related to its multi-target and multi-link action mechanism. The World Health Organization (WHO) has confirmed that traditional Chinese medicine is safe and effective in treating COVID-19. Some medicinal fungi, including *Poria cocos* and *Polyporus umbellatus*, have been widely used in clinical practice. Because of the special properties of medicinal fungi, it is better than that of traditional medicine, which is more beneficial to be used as a kind of health food. At present, the cultivation of medicinal fungi is more convenient than before, not only from the trees in the field, but also through the fermentation of mycelia. Although there are great differences in morphology between the fruiting body and the mycelium, there is little difference in the content of various components. In addition, the wide use of molecular docking technology and the development of the corresponding database bring great convenience to the screening of antiviral drugs. It can be vividly compared to adults who have changed from looking for a needle in a haystack to looking for a needle in a river. Because of the high risk of virus experiments, people can only carry out research in laboratories with biosafety levels II, III, or higher. This poses a great obstacle to the development of antiviral drugs. The use of pseudovirus solves this problem well, allowing people to avoid unnecessary risks that existed in previous experiments. On the other hand, while studying antiviral drugs, we should not be limited to the virus itself. Sometimes it is possible to grasp multiple links in which the virus replicates or invades the cell and catches a target to fight the virus. The Pfizer COVID-19-specific drug Paxlovid is aimed at the 3CL protein target of the virus. In addition to antiviral drugs, vaccines are undoubtedly another way to fight the virus. A variety of active components in medicinal fungi can not only directly fight the virus, but also activate the immune response of the human body. The use of extracts of medicinal fungi as vaccine adjuvants can significantly improve the efficacy of the vaccine to better combat viral infection. From the outbreak of highly pathogenic poultry in 1997 to Severe Acute Respiratory Syndrome (SARS) in 2003, the influenza pandemic in 2009, and today’s COVID-19, we are facing not only the spread of a virus. The co-infection caused by the combined effect of various viruses on the human body is also a problem that we should consider. The biological treasure of medicinal fungi remains to be excavated.

## Figures and Tables

**Figure 1 molecules-27-04457-f001:**
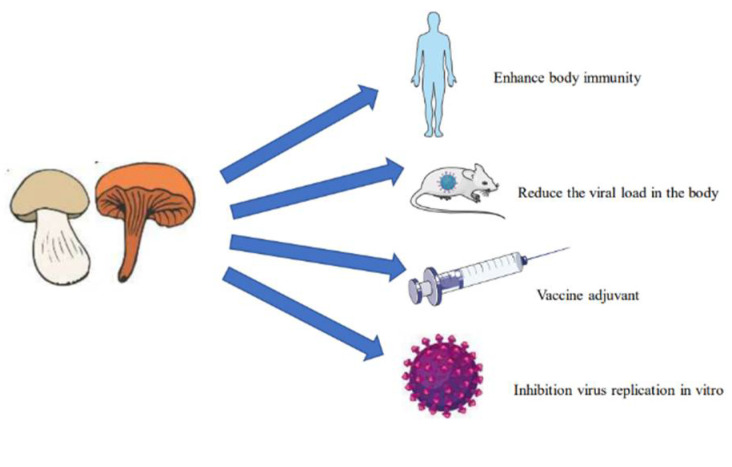
Medicinal fungi antiviral actions.

**Figure 2 molecules-27-04457-f002:**
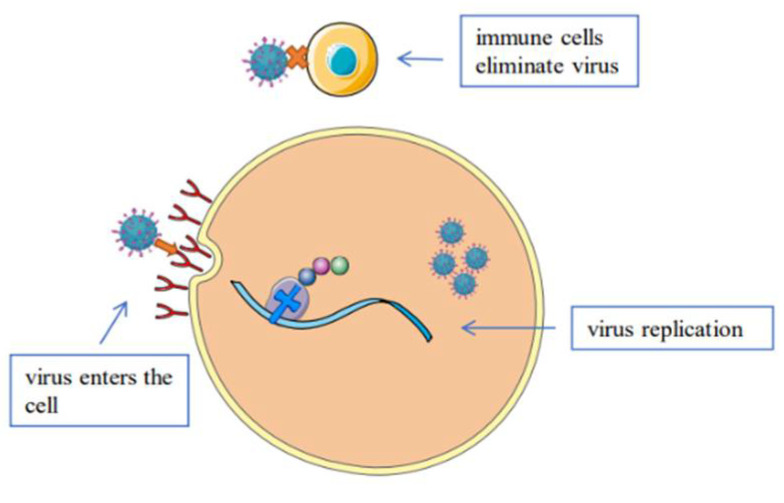
Antiviral mechanism of medicinal fungi in vivo.

**Figure 3 molecules-27-04457-f003:**
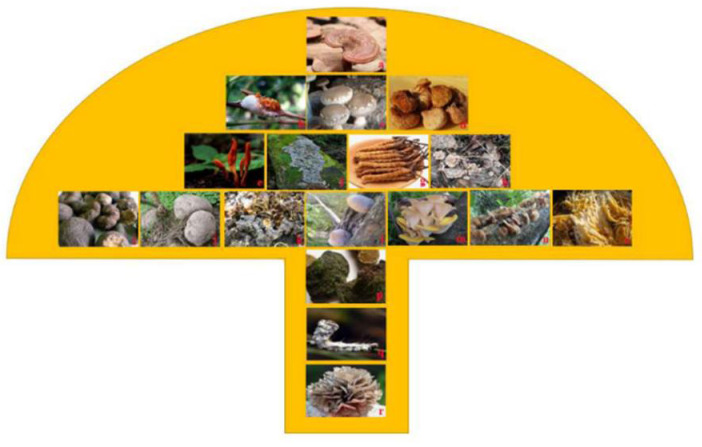
The medicinal fungi involved in this review, a–f are *Ganoderma lucidum*, *Shiraia bambusicola*, *Lentinus edodes*, *Hericium erinaceus*, *Cordyceps militaris*, *Coriolus versicolor*, *Cordyceps sinensis*, *Schizophyllum commune*, *Omphalia lapidescens*, *Poria cocos*, *Polyporus umbellatus*, *Cryptoporus volvatus*, *Pleurotus citrinopileatus*, *Auricularia auricula*, *Flammulina velutipes*, *Phellinus igniarius*, *Beauveria bassiana*, and *Grifola frondose*.

**Figure 4 molecules-27-04457-f004:**
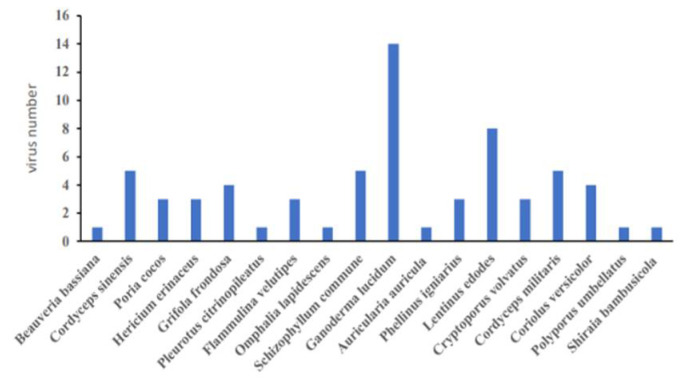
Medicinal fungi can resist the virus number in this review.

**Table 1 molecules-27-04457-t001:** Antiviral activities of medicinal fungi and their mechanism of action in human viruses.

Virus		Chemical Class	Antiviral Agent	Source	Action Mechanism	Reference
SARS-CoV-2		Polysaccharide	L-fucose–containing polysaccharides	*Ganoderma lucidum*	Vitro	[12]
		Triterpenes, protein		*Ganoderma lucidum, Grifola frondosa et al.*	Inhibition ACE activity	[13]
		Steroid	Antcins	*Antrodia cinnamomea*	ACE2 inhibitory effect	[14]
		Polysaccharide	20% α-glucans	*Lentinus edodes*	Regulating the immune system	[15]
		Polysaccharide	Carbosynth-lentinan (50%β-glucan 12%α-glucan)	*Lentinus edodes*	Reduce the level of inflammation in the lung	[16]
		Polysaccharide	In-house Lentinan (72%β-glucan)	*Lentinus edodes*	Reduce the level of inflammation in the lung	[16]
		Adenosine	Cordycepin(C_10_H_13_N_5_O_3_)	*Cordyceps sinensis*	Spike protein inhibitor	[17]
IV		Crude extract	ME, MFs (MF1, MF2 and MF3)	*Grifola frondosa*	Vitro induce the production of cytokines such as TNF-α, which inhibit the growth of virus in vitro.	[18]
		Pyranone	Phelligridins E, G	*Phellinus igniarius*	Vitro neuraminidase inhibitory activity	[19]
		Crude extract	Aqueous extract PIW	*Phellinus igniarius*	Vitro interfering with the early replication process of the virus	[20]
		Crude extract	Lentinus edodes mycelium extract LEM	*Lentinus edodes*	Vitro Inhibit virus entry into host cells activate the immune response through the type I IFN pathway	[21]
		Terpenoids	Cryptoporic acid E	*Cryptoporus volvatus*	Vitro	[22]
		Crude extract	Water extract	*Cryptoporus volvatus*	Vitro	[23]
		Polysaccharide	APS (D-galactose, L-arabinose, D-xylose, L-rhamnose, and D-galacturonic acid)	*Cordyceps militaris*	Vitro	[24]
		Crude extract		*Cordyceps militaris*	Vivo increased IL-12 expression and greater number of NK cells	[25]
		Crude extract		*Auuriporia aurea, Flammulina velutipes, Fomes fomentarius, Ganoderma lucidum, Lentinus edodes, Lyophyllum shimeji, Pleurotus eryngii, Pleurotus ostreatus, Schizophyllum commune, and Trametes versicolor*	Vitro	[26]
		Fatty acid	(2Z,4E)-deca-2,4-dienoic acid (DDEA)	*Cordyceps sinensis*	Reduce the level of inflammation caused by infection with H1N1	[27]
		Polysaccharide	PCP-II (fucose, mannose, glucose and galactose in molar ration of 1.00:1.63:0.16:6.29)	*Poria cocos*	Vaccine adjuvant	[28]
		Polysaccharide	Mixed polysaccharides (MPs)	*shiitake mushroom, poriacocos, ginger, tangerine peel*	Enhanced the level of cellular immunity and humoral immunity	[29]
EV71		Polysaccharide	Neutral polysaccharide having a β-1,6-linked Glcp backbone with 1,3-α-glucan units	*Grifola frondosa*	Vitro Inhibition of VP1 protein expression and genomic RNA synthesis	[30]
		Polysaccharide	95%polysaccharides (glucose 79% and mannose 21%) and 5% proteins	*Ganoderma lucidum*	Vaccine adjuvant	[31]
		Triterpenoid	Lanosta-7,9(11),24-trien-3one,15;26-dihydroxy (GLTA) and Ganoderic acid Y (GLTB)	*Ganoderma lucidum*	Vitro inhibits the replication of the viral RNA replication through blocking EV71 uncoating	[32]
HIV		Adenosine	Cordycepin(C_10_H_13_N_5_O_3_)	*Cordyceps sinensis*	Vitro inhibitory activity on HIV-1 reverse transcriptase IC50 8.2 × 10^−3^ μM	[33]
		Water extract		*Cordyceps sinensis*	Vitro inhibitory activity on HIV-1 reverse transcriptase	[34]
		Adenosine	L3a(C_10_H_13_N_5_O_4_), L3b(C_20_H_20_O_7_), L3c(C_12_H_17_N_5_O_5_)	*Cordyceps militaris*	Vitro inhibitory activity on HIV-1 reverse transcriptase	[35]
		Lectin	N-terminal amino acid sequence (NSTDISLNHG) Molecular mass 30 kDa	*Cordyceps militaris*	Vitro inhibitory activity on HIV-1 reverse transcriptase IC50 10 μM	[36]
		Lectin	N-terminal amino acid sequence (QYSQMAQVME) Molecular mass 32.4 kDa	*Pleurotus ostreatus*	Vitro inhibitory activity on HIV-1 reverse transcriptase IC50 0.93 μM	[37]
		Protein	N-terminal amino acid sequence (AEGTLLGSRA TCESGNSMY) Molecular mass 9567 Da,	*Hypsizigus marmoreus*	Vitro inhibitory activity on HIV-1 reverse transcriptase IC50 30 nM	[38]
		Protein	N-terminal amino acid sequence(XHPDLFXX) molecular mass 13.8 kDa	*Flammulina velutipes*	Vitro inhibitory activity on HIV-1 reverse transcriptase	[39]
		Hemolysin	N-terminal amino acid sequence (ATNYNKCPGA) Molecular mass 29 kDa	*Schizophyllum commune*	Vitro inhibitory activity on HIV-1 reverse transcriptase IC50 1.8 mM	[40]
		Ribonuclease	N-terminal amino acid sequence (TPYLDYLAAL QADGPVVPFIRNWEGALSIS) Molecular mass 20 kDa	*Schizophyllum commune*	Vitro inhibitory activity on HIV-1 reverse transcriptase IC50 65 mM	[41]
		Crude extract	NGCs and AGCs	*Ganoderma lucidum*	Vitro inhibitory effects on the attachment of HIV-1 glycoprotein 120 to cluster of differentiation 4	[42]
		Triterpenoid	Ganoderic acid B	*Ganoderma lucidum*	Vitro inhibitory activity on HIV-1 reverse transcriptase	[43]
		Triterpenoid	Ganoderic acid β, lucidumol B, ganodermanondiol, ganodermanontriol, ganolucidic acid A	*Ganoderma lucidum*	Vitro inhibitory activity on HIV-1 reverse transcriptase IC50 20, 50, 90, 70, 70 μM	[44]
		Triterpenoid	Ganoderiol F and ganodermanontriol	*Ganoderma lucidum*	Vitro anti-hiv-1 activity with an inhibition concentration of 7.8 μg/mL	[45]
		Triterpenoid	Ganoderic acid B, ganoderiol B, ganoderic acid C1, 3b-5a-dihydroxy-6b-methoxyergosta-7,22-diene, ganoderic acid a, ganoderic acid H and ganoderiol A	*Ganoderma lucidum*	Vitro inhibitory activity on HIV-1 reverse transcriptase IC50 0.17–0.23 mM	[45]
		Polysaccharide	PSP (28% polysaccharide-to-peptide ratio and a composition of 60.23 mg/g beta-1,3/1,6-glucan)	*Coriolus versicolor*	Vitro inhibitory activity on HIV-1 reverse transcriptase IC50 6.25 μg/mL inhibition of the interaction between HIV-l gp120 and immobilized CD4 receptor	[46,47]
		Laccase	N-terminal amino acid sequence (AGTSHFADL) molecular mass 67 kDa	*Lentinus edodes*	Vitro inhibitory activity on HIV-1 reverse transcriptase IC50 7.5 µM	[48]
		Protein	N-terminal amino acid sequence (CQRAFNNPRDDAIRW) molecular mass 27.5 kDa	*Lentinus edodes*	Vitro inhibitory activity on HIV-1 reverse transcriptase IC50 1.5 µM	[49]
		Lignin	The elemental analysis; C, 44.6%; H, 4.68%; N, 1.74%	*Lentinus edodes*	Vitro	[50]
		Water extract	E-P-LEM and LEM	*Lentinus edodes*	Vitro	[51]
HPV		Crude extract	Fruit body powders	*Trametes versicolor, Ganoderma lucidum*	Vivo	[52]
		Crude extract	Non-hormonal gel	*Trametes versicolor*	Vivo	[53]
DENV		Conidia	Conidia	*Beauveria bassiana*	Vivo activation of the mosquito’s anti-dengue Toll and JAK-STAT pathways	[54]
		Adenosine	Cordycepin(C_10_H_13_N_5_O_3_)	*Cordyceps sinensis*	Vitro inhibition of viral RNA replication	[55]
		Crude extract	Hot aqueous extracts, and aqueous soluble extracts	*rhinocerotis*, *Pleurotus giganteus*, *Hericium erinaceus*, *Schizophyllum commune*	Vitro	[56,57]
		Aqueous extract	Hesperidin	*Ganoderma lucidum*	Vitro anti-DENV NS2B-NS3 protease activity	[58]
		Triterpene	Ganodermanontriol	*Ganoderma lucidum*	Vitro	[59]
HV	HBV	Crude extract	Crude extract	*Cordyceps sinensis*	Reduce apoptosis of renal tubular epithelial cells	[60]
		Crude extract	GF-D extract	*Grifola frondosa*	Vitro directly interferes with HBV replication at the level of DNA polymerase	[61]
		Polysaccharide	β-glucan with a (1-3)-β-glucose backbone and (1-6)-β-glucose side chains	*Polyporus umbellatus*	Treat hepatitis caused by HBV	[62]
		Polysaccharide	Glucan	*Schizophyllum commune*	Modulate both cellular and humoral immune responses	[63]
		Crude extract	Co-fermented broth	*Ganoderma lucidum and Radix Sophorae flavescentis*	Vitro	[64]
		Triterpene	Ganoderic	*Ganoderma lucidum*	Vitro	[65]
		Polysaccharide	LEP-1 (glucose72.15%, galactose12.12%, mannose10.02%) LEP-2(glucose79.49%, galactose7.12%, mannose6.92%)	*Lentinus edodes*	Vitro	[66]
	HCV	Crude extract	MSCE and LM-lignin	*Lentinula edodes*	Inhibit the entry of HCVpv into cells	[67]
		Adenosine	Cordycepin(C_10_H_13_N_5_O_3_)	*Cordyceps militaris*	Inhibit HCV RNA replication in vitro inhibit NS5B polymerase activity	[68]
HV	HSV	Protein	N-terminal amino acid sequence (NH2-REQDNAPCGLN-COOH) molecular mass 29.5 kDa	*Grifola frondosa*	Vitro and vivo	[69]
		Crude extract	JLS-SO01	*Lentinus edodes*	Vitro blocks HSV-I replication at the later stage of the virus replication cycle	[70]
		Polysaccharide	APBP polysacchride (approximately 40.6%) and protein (approximately 7.80%) carbohydrates molar ratio (C:H:O = 1:2:1)	*Ganoderma lucidum*	Vitro	[71,72,73]
		Crude extract	Water-soluble extracts GLhw and GLlw and eight kinds of methanol-soluble extracts GLMe-1-8	*Ganoderma lucidum*	Vitro	[74]
		Crude extract	Herbal mixture WTTCGE	*Ganoderma lucidum*	Vivo	[75]
	EBV	Adenosine	Cordycepin(C_10_H_13_N_5_O_3_)	*Cordyceps sinensis*	Vitro Affect the synthesis of viral proteins by acting on the genes of EBV	[76]
		Terpenoid	lucidenic acid P, methyl lucidenates P, methyl lucidenates Q	*Ganoderma lucidum*	Inhibitory effects on Epstein-Barr virus early antigen induction	[77]
		Terpenoid	Ganoderic acid A, ganoderic acid B, ganoderol B, ganodermanontriol, ganodermanondiol	*Ganoderma lucidum*	Vitro	[78]
RSV		Protein	114 amino acid residues molecular mass 13 kDa	*Flammulina velutipes*	In vivo reduce the viral titers of RSV reducing NF-jB translocation	[79]
PV		Crude extract polysaccharide	LeP(β-D-glucan) aqueous (AqE) and ethanol (EtOHE) extracts	*Lentinus edodes*	Vitro	[80]
RV		Polysaccharide	Fucose, mannose, glucose and galactose at a molar ratio 1.00:1.63:0.16:6.29.	*Poria cocos*	Adjuvant for rabies vaccine	[81]
MARV		Polysaccharide	PCP-II (fucose, mannose, glucose and galactose in molar ration of 1.00:1.63:0.16:6.29)	*Poria cocos*	Vaccine adjuvant	[82,83]

**Table 2 molecules-27-04457-t002:** Antiviral activities of medicinal fungi and their mechanism of action in plant and animal viruses.

Virus		Chemical Class	Antiviral Agent	Source	Action Mechanism	Reference
Animal viruses	IHNV	Polysaccharide	LNT-I β-(1→3)-glucan backbone with -(1→6)-glucosyl side-branching units glucose, mannose and galactose with the molar ratio of 19.26:1.20:1.00	*Lentinus edodes*	Vitro directly inactivate virus and inhibit virus replication	[84]
	MDRV	Polysaccharide	HEP glucose (51.02%), galactose (42.24%), mannose (4.5%) and arabinose (2.2%)	*Hericium Erinaceus*	Vivo improve the number of intestinal mucosal immune-related cells regulate the homing process of muscovy duck lymphocytes	[85,86,87]
	WSSV	Polysaccharide	β-1,3 glucan BG	*Schizophyllum commune*	Enhance the immune level of shrimp	[88]
	FIV	Crude extract	Ethanol extract	*Inonotus obliquus*	Vitro FIV reverse transcriptase inhibitory effect	[89]
		Crude extract	Commercially available compound HELP-TH1	*Ganoderma lucidum, Cordyceps sinensis, Trametes versicolor*	Vitro	[90]
	DWV LSV	Crude extract		*Fomes fomentarius, Ganoderma applanatum*	Vivo	[91]
	NNV	Protein	Recombined protein rLZ-8	*Ganoderma lucidum*	Vivo activate the immune defense	[92]
	PCV-2	Polysaccharide	PS (D-glucose, D-xylose, D-galactose, L-fucose, D-mannose, and L-rhamnose at a molar ratio of 5.35:2.67:1:1.19:0.38:0.37)	*Ganoderma lucidum*	Vaccine adjuvant	[93]
	PRRSV		C_M-H-L-5_	*Cryptoporus volvatus*	Vitro	[94]
		Ergosterol	5α,8α-epidioxy-22E-ergosta6,22-dien-3β-ol	*Cryptoporus volvatus*	Vitro	[95]
		Crude extract	Water extract	*Cryptoporus volvatus*	Vitro and vivo inhibited the entry of PRRSV and the synthesis of PRRSV RNA	[96]
	PDCoV	Triterpene	Ergosterol peroxide EP	*Cryptoporus volvatus*	Vitro and vivo activate p38/MAPK and NF- κB signal pathways	[97,98]
	BoHV-1	Crude extract polysaccharide	LeP(β-D-glucan) aqueous (AqE) and ethanol (EtOHE) extracts	*Lentinus edodes*	Vitro	[80]
	NDV	Crude extract	Methanol and n-butanol fractions	*Ganoderma lucidum*	Vitro antineuraminidase activity	[99]
		Polysaccharide	GLP	*Ganoderma lucidum*	Vaccine adjuvant	[100]
		Polysaccharide	AAP and sulfated polysaccharide sAAP	*Auricularia auricula*	Vitro	[101]
		Polysaccharide	CMP40, CMP50	*Cordyceps militaris*	Vaccine adjuvant	[102]
Plant viruses	TMV	Steroid	Leiwansterols A, B	*Omphalia lapidescens*	Vitro	[103]
		Polysaccharide	BAS-F acidic polysaccharide contain carbon and hydrogen	*Fomes fomentarius*	Completely inhibit leaf infection and no toxic effect	[104]
		Polysaccharide	LNT β-(1→3)-linked backbone of d-glucose residues, two β-(1→6)-d-glucosyl residues are attached for everyfive main-chain d-glucose residues	*Lentinus edodes*	Vitro affects the affinity of TMV coat protein to host and activate some defense genes	[105]
	GBNV	Crude extract	Mixed culture filtrate	*Coprinopsis cinerea*, *Ganoderma lucidum*, *Lentinula edodes*	Vitro	[106]

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
