# Peer review of "Medicinal Fungi with Antiviral Effect"

_molecules, 2022, doi:10.3390/molecules27144457_

Round 1
Reviewer 1 Report
Dear Authors,
Thank you for making attempt of reviewing many manuscripts and research and trying to consolidate them in the one comprehensive review article. It is not easy task, epsecially in terms of medical related topics. Especialy for your consideration of animal and plant diseases and possibilities ti use fungi derived compounds for the treating, this is the first of kind review I read which is related to animals and plants.
Native speaker should be invited for the proofreading and editing of the manuscript. Many spelling and grammar mistakes.
I would very carefully used word DRUG in the context of substances delivered from mushrooms, because for my best knoladge there are not established drugs from higher fungi. Some substances are potentialy being reported to have some therapeutic effects, however no drugs have been registered. I would recommand to use "substances with therapeutic, with antiviral effect or substances which cause some sort of other antiviral effect". Because DRUG is a registered substance of known, established and wel reserched action.
Author Response
Point 1: Native speaker should be invited for the proofreading and editing of the manuscript. The article has many spelling and grammar mistakes.
Response 1: We have invited a native English-speaking colleague to review and revise our manuscript.
Point 2: I would very carefully used word DRUG in the context of substances delivered from mushrooms, because for my best knoladge there are not established drugs from higher fungi. Some substances are potentialy being reported to have some therapeutic effects, however no drugs have been registered. I would recommand to use "substances with therapeutic, with antiviral effect or substances which cause some sort of other antiviral effect". Because DRUG is a registered substance of known, established and wel reserched action.
Response 2: We have replaced the word DRUG with agents and substances. (Page 3 line 92 and 100)
Reviewer 2 Report
The article is written according to the classical scheme for reviews and consists of an introduction, a description of the data obtained in recent years on the antiviral activity of substances isolated from fungi and a conclusion.
Despite the generally positive impressions of the article, the reviewer has a number of comments:
1- In sections 2.1 and 2.2, it is necessary to replace the word "virus" with "viruses", since many viruses are listed further in the text. Similarly in paragraph 2.1.7 and 2.3
2- Section 2.2. Animal viruses is mistakenly numbered as 2.1. This needs to be fixed.
Also, at the end of each section, the authors are kindly requested to provide a brief overview of those fungal origin compounds that have reached the stage of using in medical, veterinary or agriculture practice.
Author Response
Point 1: In sections 2.1 and 2.2, it is necessary to replace the word "virus" with "viruses", since many viruses are listed further in the text. Similarly in paragraph 2.1.7 and 2.3
Response 1: We have replaced the word "virus" with "viruses" in sections 2.1, 2.1.7, 2.2, and 2.3.
Point 2: Section 2.2. Animal viruses is mistakenly numbered as 2.1. This needs to be fixed.
Response 2: We have fixed the section number in section 2.2.
Point 3: At the end of each section, the authors are kindly requested to provide a brief overview of those fungal origin compounds that have reached the stage of using in medical, veterinary or agriculture practice.
Response 3: We have supplemented a brief overview of those fungal origin compounds that have reached the stage of using in medical, veterinary or agriculture practice. (Page 10 line 428 to 441, Page 18 line 567 to 579, 603 to 609)